# PROCEEDINGS A

# Research

graph theory, computer modelling and simulation, human-computer interaction

social coordination under uncertainty, collective intelligence, information diffusion, communication network, social experiments

**Author for correspondence:**
Hirokazu Shirado
e-mail: shirado@cmu.edu

One contribution to a special feature 'A generation of network science'.

# Collective communication and behaviour in response to uncertain 'Danger' in network experiments

Hirokazu Shirado[1], Forrest W. Crawford[2,4] and Nicholas A. Christakis[2,3]

[1]School of Computer Science, Carnegie Mellon University, Pittsburgh, PA 15213, USA
[2]Yale Institute for Network Science, and [3]Department of Sociology, Yale University, New Haven, CT 06520, USA
[4]Department of Biostatistics, Yale School of Public Health, New Haven, CT 06510, USA

HS, 0000-0003-4545-7859

In emergencies, social coordination is especially challenging. People connected with each other may respond better or worse to an uncertain danger than isolated individuals. We performed experiments involving a novel scenario simulating an unpredictable situation faced by a group in which 2480 subjects in 108 groups had to both communicate information and decide whether to 'evacuate'. We manipulated the permissible sorts of interpersonal communication and varied group topology and size. Compared to groups of isolated individuals, we find that communication networks suppress necessary evacuations because of the spontaneous and diffuse emergence of false reassurance; yet, communication networks also restrain unnecessary evacuations in situations without disasters. At the individual level, subjects have thresholds for responding to social information that are sensitive to the negativity, but not the actual accuracy, of the signals being transmitted. Social networks can function poorly as pathways for inconvenient truths that people would rather ignore.

# 1. Introduction

Collective dangers—including epidemics [1,2], economic crises, and natural and human-caused disasters [3–5]—pose a grave challenge to human coordination and communication. For example, when a disaster occurs, prompt and reliable information exchange, coordinated behaviour and self-sacrifice all play a role in individual and collective safety [6]. Although coordination is particularly important in such situations, many network phenomena—such as the spread of false rumours and social disconnection [7] —may jeopardize people's well-being, especially as online communication becomes increasingly important as a means of coordination [8–10]. Injuries and deaths related to human responses to disasters can often exceed the direct impact of the disaster itself [3].

Some of the coordination difficulty comes from the asymmetrical behavioural consequences associated with uncertainty [11]. In a threatening situation, the payoffs that people experience depend on whether people have accurate information and on how they act, and also on whether and how other people communicate and act. If nothing will happen, people like to stick with the status quo—because taking protective actions involves economic and psychological costs [12]. However, when people do not act and an actual danger materializes, they can suffer large losses at both individual and collective levels [4]. Information gathering is often critical to making accurate judgements [13], and misinformation can cause actual damage due to procrastination and misjudgement. In addition, various dilemmas can also arise; for instance, taking time to be correct collectively (i.e. by staying longer in order to pass on information, or by taking the time to collect more accurate information) can increase an individual's risk of being adversely affected.

Given such factors, it is unclear whether people connected with each other respond better to an uncertain danger than a similar number of isolated individuals. While social networks are often seen as reliable information pipelines [14], they may also magnify individuals' bias and uncertainty [15–18]. Theory suggests that informational cascade can occur irrespective of whether the information is right or wrong [19,20]. Uncertain danger can cause two types of errors in people; on the one hand, the spread of false alerts could trigger a chain reaction of unnecessary confusion (in a kind of type I error) [8,9]; on the other hand, people could procrastinate rather than make a necessary evacuation in the face of an impending disaster and that they promulgate false reassurance, known as 'normalcy bias', in emergencies (hence, a kind of type II error) [21,22]. It is likely that interpersonal communication suppresses one type of error, but amplifies the other [23].

Here, we evaluate these ideas using laboratory experiments involving real economic stakes and a networked decision-making scenario simulating an unpredictable and sudden 'disaster'. We focus on the interplay between interpersonal communication (regarding what participants indicate is happening) and behavioural decisions (regarding whether to 'evacuate'), which is critical in this type of situation [6,24]. Furthermore, this interplay between communication and action may depend on the structure of social networks [25–28]. Hence, in our primary experiment, we manipulate network topology and size. Our approach supplements observational field studies [5], and our experiments provide systematic measurement of social contagion both when an uncertain danger materializes and when it does *not*. Within this randomized controlled setting, we also explore the spontaneous emergence of true and false information about the 'danger' and the impact of the propagation of this information on individual and collective behaviour.

# 2. Experiment set-up

We recruited 2480 unique subjects via an online labour market and randomly assigned them to various conditions in which subjects could communicate with each other (in $N = 108$ groups). We also assigned 168 individuals to an *independent* condition affording no interpersonal communication. And we conducted a number of supplementary experiments involving a further 1700 subjects in 93 groups. Overall, 4348 people participated in our experiments.

**Table 1.** Player's payoffs. The value of each cell is a payoff (US$) that a subject would receive. In a session, subjects receive a US$2 endowment at the outset. When they evacuate to avoid a possible disaster, they need to pay US$1. If a disaster strikes and subjects have not evacuated, they lose their entire endowment. Otherwise, they receive US$0.10 per other player in their group who has chosen a correct behaviour (either evacuate or stay), in addition to their own leftover endowment.

| | disaster occurs | |
| --- | --- | --- |
| | yes | no |
| player's choice | | |
| evacuate | $1.0 + 0.1 \times n_{Evacuate}$ | $1.0 + 0.1 \times n_{Stay}$ |
| | (true positives) | (false positives) |
| stay | 0.0 | $2.0 + 0.1 \times n_{Stay}$ |
| | (false negatives) | (true negatives) |

As in many other empirical studies [11,12,18,29–32], we used small economic stakes to examine social coordination and risk-based decision-making. Subjects played a decision-making game in which the goal is to make an appropriate decision regarding whether to evacuate from an impending 'disaster' that would wipe out subjects' endowment unless they evacuated in time. Subjects received US$2.00 at the outset. If nothing happened until the short (75-second) game suddenly ended, they kept the endowment. However, they might be involved in a 'disaster' that could strike at any second. Each subject could spend US$1.00 at any time to leave the game and avoid this danger. Evacuated subjects reduced their endowment even when nothing happened. In addition, when subjects were not involved in a 'disaster' (i.e. either when a disaster does not materialize or when they successfully evacuate before it materializes), they earned US$0.10 for every other player who took the correct action (table 1). This additional payment simulates positive externalities in one's community (e.g. future socioeconomic benefits from the well-being of others) and serves to incentivize the communication of reliable information about the 'disaster'. The payoff architecture shown in table 1 presents every subject with a dilemma similar to that faced in emergency decision-making as was described above (though, of course, this is a stylized game). Subjects were incentivized to *stay* due to the evacuation cost, but also incentivized to be *correct*. The worst scenario was to *stay* but be *incorrect*. And subjects were also incentivized to communicate to others as accurately as possible. Subjects could communicate (except for the independent condition; see below) and make their behavioural decisions using software buttons in a game window (see electronic supplementary material).

In the social network conditions, subjects who had not yet evacuated could share their view about the likelihood of an impending 'disaster' with their network 'neighbours'. Subjects ($N = 2480$) were randomly assigned to a location in a network in each of 108 groups where they all had four connections (figure 1a). At the start of the game, one randomly selected subject in each group (the 'informant') was told in advance whether a 'disaster' would indeed strike. This simulates the existence of true, accurate knowledge, initially possessed by a small number of people such as experts or eyewitnesses (e.g. those closer to an approaching wildfire) [9]. The other subjects were informed that some players indeed had accurate information about the 'disaster', and they were also told that immediate neighbours of the informant (always $N = 4$) would know the identity of the true informant. As is often the case in real situations where groups face a sudden and uncertain danger, most subjects have no independent way of distinguishing fact from opinion; subjects had to resort to social cues and private preferences in order to make decisions about what to say or do.

In the network conditions, subjects were allowed to share information about the possibly impending 'disaster' by using 'Safe' and 'Danger' buttons that indicated their assessment. When they clicked the Safe button, their node turned blue and, after 5 s, automatically returned to grey. Likewise, the Danger button turned their node red for 5 s. Subjects could see only the colours of neighbours to whom they were directly connected (see electronic supplementary

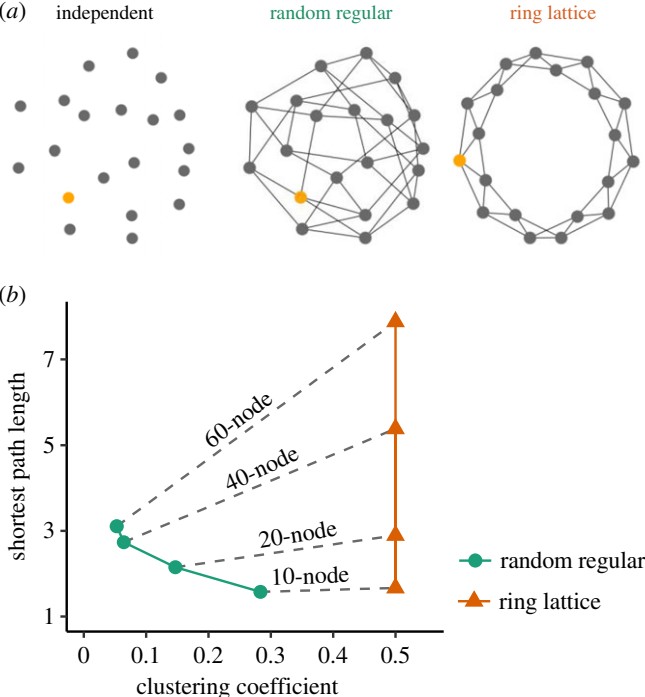

**Figure 1.** Experiment condition. (*a*) Network topology with 20 nodes. Each session has one informant per 20 subjects (indicated by a yellow node). In the sessions of both random-regular and ring-lattice networks, subjects have exactly four neighbours to communicate with. In the sessions without a network (i.e. the independent condition), subjects are isolated from each other. (*b*) Network properties vary with network topology and size. Random regular in green with circles, ring lattice in orange with triangles. (Online version in colour.)

material). Once subjects chose to evacuate, they could no longer send signals, and their node showed grey (the default colour) for the rest of the game. The neighbours of evacuated subjects were not informed of their evacuation. This basic setting simulates the situation of people who simply lose communication with each other during an evacuation. We also tested other settings about evacuees' communication ability and behavioural visibility in supplementary experiments (see below).

Within this basic set-up, we manipulated network topology and size. Subjects were randomly assigned to one of two network types: a random-regular network and a ring-lattice network (figure 1*a*). While the local environment was the same (i.e. all the subjects had four neighbours), the global topology varied substantially. Ring-lattice networks have a high level of clustering created by redundant ties and also a longer geodesic distance between any two nodes, on average (figure 1*b*). As a network increases in size, the difference in topology between random-regular and ring-lattice networks becomes larger (figure 1*b*). The properties of real-world networks are typically in between these two extremes [25,26]. We also tested two further types of network structures in supplementary experiments.

Independent of network topology, we manipulated network size to have 10, 20, 40 or 60 nodes. Since each network had only one informant, the proportion of informed individuals varied; while 10% of the subjects were given accurate 'disaster' information in 10-node networks, only 1.5% knew it in advance in 60-node networks. Subjects were not informed of the network type and size to which they were assigned, and they all had the same degree (i.e. four connections), regardless of the network they were assigned to.

In the independent condition, subjects ($N = 168$) played the game without social interactions (figure 1*a*). We informed eight subjects of a 'disaster' striking and eight subjects of a 'disaster'

not striking, and 152 subjects across the sessions knew nothing. Overall, the portion of informed and uninformed individuals in the independent sessions is equivalent to that in the sessions of 20-node network (5% of 160 subjects were informed about the 'disaster' status for each treatment). These games had the same payoff architecture and user interface as the network conditions; in the independent condition, however, they could not exchange information because they were not connected with each other (figure 1*a*).

As noted, all the sessions in all conditions ended suddenly at 75 s. In precisely half the sessions (i.e. with the probability of 50%), we contrived to have a 'disaster' suddenly strike at the end of a game, at the 75-second mark. Subjects were merely informed during the game that 'A disaster may or may not strike'. We did not inform any subjects (including the informants) about *when* the session would end. Thus, even individuals who were informants could fail to escape while they stayed in the game to help others by sending signals [29].

We confirmed that subjects understood the rules of the game with: extensive pilot testing of our interface; a screening test of understanding of game rules and payoffs before subjects could play the game (see Material and methods) and debriefing focus groups to ask people why they played as they did (see below). Subjects understood the rules but still often made 'irrational' choices. Subjects could participate in the experiment only once.

In summary, subjects could do two things: spread information about their impressions (whether true or false) of the safety or danger of the situation and decide whether to actually take an action (by 'evacuating' from the game). We conducted 108 network sessions with 2480 subjects for the network conditions: 48 networks with 10 subjects, 32 networks with 20 subjects, 16 networks with 40 subjects and 12 networks with 60 subjects. Half of the networks were random-regular and half were ring-lattice. In both conditions, half of the sessions were set-up so that a 'disaster' eventually struck and half were 'disaster'-free. And we examined the independent condition with 168 subjects (i.e. 168 solo sessions).

In our scenario, interpersonal communication and reward-related decision-making are *decoupled* [33]. Signal selection *per se* does not directly affect a subject's payoff (table 1); it produces value only with the decision whether to evacuate and with respect to what others choose to do in response to the signal. Also, subjects do not need to match their words with their actions in this game. Thus, this game allowed us to explore the spontaneous emergence of true and false information about an impending danger [6,8,9] and the impact of the propagation of this information across the network, as well as how subjects cope with possibly conflicting information in their social environment.

## 3. Results

## (a) Collective behaviour in response to an uncertain 'danger'

In the independent condition with no social interactions, 65 out of 152 uninformed subjects (that is, excluding the 16 informed subjects) evacuated in less than the allotted 75 s. With the payoff structure shown in table 1, the expected payoff from evacuation is higher than that of staying under an even chance of a 'disaster' (see Material and methods). Thus, in this situation, 57.2% of uninformed subjects did not evacuate, contrary to the principle of economic rationality under the assumptions here.

We therefore evaluated whether social communication interactions affected (and might ameliorate) behavioural biases in responding to a possible imminent 'disaster'. Figure 2 shows evacuation decision-making in the sessions involving the eight network conditions, compared to that of the independent condition (see electronic supplementary material, figure S1 for detailed results). Before implementing pairwise comparisons, we performed a log-rank test of the null hypothesis that all the curves in figure 2 are identical; that hypothesis was rejected ($p < 0.001$), indicating at least two of the survival curves differed. The foregoing results persist despite a Bonferroni-type correction implemented in our analysis. We also tested the differences using Cox

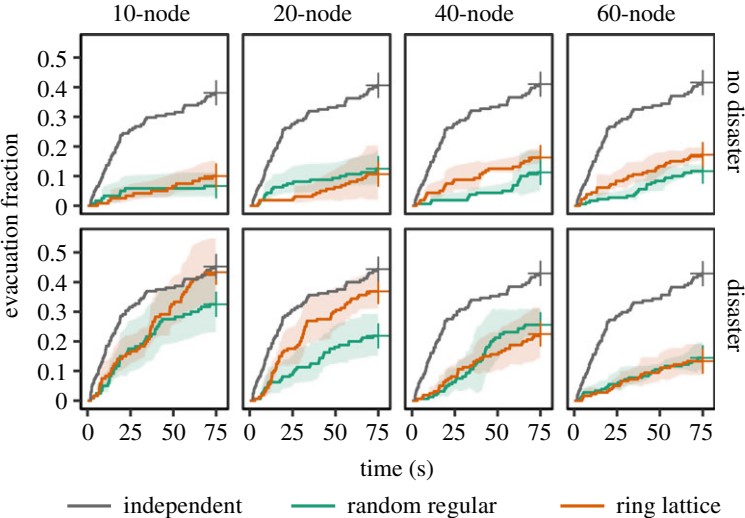

**Figure 2.** Aggregated evacuation fraction over time. Lines indicate average fractions of evacuated subjects over time in different experimental conditions. Shades are 95% confidence intervals among network sessions ($N = 12$ for 10-node; $N = 8$ for 20-node; $N = 4$ for 40-node; $N = 3$ for 60-node networks). The lines of the 'independent' condition (of solitary nodes) are identical with the exception of the single informed subject (who differed in the disaster and no-disaster situations). When a 'disaster' does *not* strike, subjects embedded in networks are *more* likely to select an appropriate action (i.e. to stay) than those subjects in the isolated condition. However, when a 'disaster' does strike, subjects embedded in networks are *less* likely to select an appropriate action (i.e. to evacuate) and more likely to suffer the consequences of the disaster than those in the independent condition. Random regular in green; ring lattice in orange. (Online version in colour.)

proportional hazards models incorporating a random effect for sessions and found similar results (see electronic supplementary material, table S1).

When a 'disaster' would *not* strike, subjects successfully stayed in the game when embedded in a network, and this was significantly better than subjects lacking social interactions across all the topology and size treatments ($p < 0.001$ for all the combinations of network topology and size; log-rank test). That is, social interactions (of any type) helped—but only when a 'disaster' did not strike. There is no meaningful difference with respect to network topology ($p = 0.378$ for 10-node networks, $p = 0.565$ for 20-node, $p = 0.170$ for 40-node and $p = 0.121$ for 60-node; log-rank test).

By contrast, when a 'disaster' did actually strike, subjects embedded in a network evacuated *less* than those without network interactions (figure 2), even though an informant, working with others, could pass on the truth about an impending calamity. That is, social interactions generally did not help subjects to avoid danger. Only ring-lattice networks in small populations reached the same level as the independent condition ($p = 0.512$ for the 10-node lattice, $p = 0.091$ for 20-node; log-rank test); the comparison was different for the random-regular networks ($p = 0.025$ for the 10-node network and $p < 0.001$ for the other networks; log-rank test). But no network condition out-performed the independent condition in terms of evacuation in situations where the 'disaster' struck.

Furthermore, in a 'disaster' situation, network structure had a significant impact on the dynamics of evacuation diffusion when the network is small. Ring-lattice networks (with greater clustering and transitivity) were significantly more effective for spreading evacuation behaviour than the random networks, especially when they consisted of 20 nodes ($p = 0.120$ for the 10-node networks, $p = 0.003$ for 20-node, $p = 0.536$ for 40-node and $p = 0.766$ for 60-node; log-rank test).

The actions taken by the groups can also be thought of as a collective prediction, with a binary classifier (see electronic supplementary material, figure S2). In the independent condition, the accuracy rate in response to a 'disaster' is the same as a random guess at the end of a session (accuracy at $75\,\text{s} = 0.519$). But social interaction and information flow within networks,

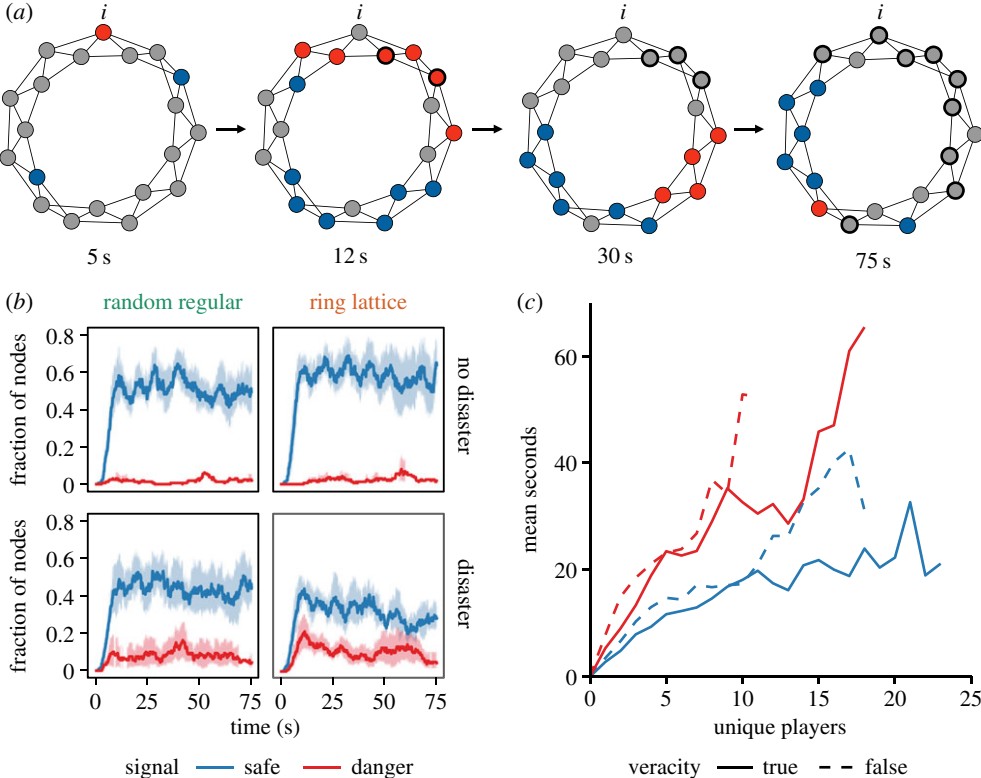

**Figure 3.** Safe signals exceed danger signals, whether a 'disaster' strikes or not. (*a*) An example of the spreading of signals in the game. The figures are snapshots at 5, 12, 30 and 75 s in a 20-node ring-lattice network with a 'disaster' striking at the end (see movie S2 for full animation). Each node's colour shows the signal choice made by subjects at the indicated time (blue for 'Safe' and red for 'Danger'). The label 'i' indicates the informant's node. Bold nodes indicate subjects who have evacuated at the time, but, in the actual game, neighbours were not informed of their evacuation. In this session, nine of 20 subjects successfully evacuated before the disaster struck. (*b*) Average fraction of nodes showing safe signals (blue lines) and danger signals (red lines) over time in 20-node networks (including evacuated subjects in the denominator) (see electronic supplementary material, figure S4 for networks of other sizes). Shading indicates 95% confidence intervals among network sessions (*N* = 8). (*c*) Speed of spread of information, by signal type and accuracy. The curves indicate the number of seconds it takes for signal cascades to reach any number of unique players in a network in all of the experimental network sizes combined. While the speed varies with signal *type* (*p* < 0.01; Kolmogorov–Smirnov test), with safe signals spreading faster, there is no detectable difference by signal *accuracy* (*p* = 0.26; Kolmogorov–Smirnov test). Safe signals in blue, danger signals in red. (Online version in colour.)

especially small networks, help subjects to increase the overall accuracy of the 'disaster' forecast by reducing false evacuation significantly (e.g. accuracy at 75 s = 0.547 with random-regular networks and 0.631 with ring-lattice networks with 20 nodes). The prediction accuracy improves with increasing clustering coefficients and decreasing average shortest path in the networks. However, the subjects within networks were *slower* in making decisions, compared to those without networks (average evacuation time among evacuees = 30.1 s (random-regular with 20 nodes) and 31.9 s (ring-lattice with 20 nodes) versus 23.4 s (independent)). Networks fostered interpersonal communications that improve the overall prediction ability of a group, but, at the same time, put subjects at risk for a substantial loss in a time-critical situation by delaying their action.

In the social network conditions, the informants did actually have a positive impact on certain other subjects. The closer to an informant a subject was, the more likely he or she evacuated (figure 3*a*; see electronic supplementary material, figure S3 for the aggregated results).

However, the baseline likelihood of evacuation was quite low (less than 20% at 75 s) in the subjects embedded in a network (compared to isolated players). In short, an informant's effect was not strong enough to make indirectly connected subjects evacuate at the same level as the independent condition, especially in a large network.

## (b) Signal diffusion in communication networks

Turning to the dynamics of information diffusion, we find that the spreading of true or false reassurance overwhelms that of warnings, regardless of network topology and evacuee's connectivity (figure 3*a*). Within 10 s after the onset of a networked session with 20 nodes, safe signals (most of which were actually *false*) spread over about 40% of the population with an informant informed of the impending 'disaster', and over about 60% with an informant informed of no impending 'disaster'. This fraction of safe-signalling nodes is maintained virtually constant after the initial surge. The diffusion level of danger signals is almost always less than that of safe signals (figure 3*b*). Different network types and sizes show the same pattern in the dynamics of information diffusion (see electronic supplementary material, figure S4).

Indeed, the empirically observed diffusion speed of the signal depends on the signal's type (safe or danger) rather than the signal's accuracy (true or false) (figure 3*c*; see Material and methods for the calculation procedure). We found that, while safe signals spread significantly faster than danger signals ($p < 0.001$; Kolmogorov–Smirnov test), the diffusion speed has no meaningful difference with respect to the signal's accuracy ($p = 0.258$; Kolmogorov–Smirnov test). As a result, false reassurances spread faster than true warnings (figure 3*c*).

To gain insight into the foregoing behaviours, we elicited observations from about 160 people involved in additional sessions of people who played this game in a classroom setting. These people participated in a kind of focus group, involving roughly 20–40 people in a large room playing the games together. We did not numerically quantify responses, but we used subjects' observations to get a qualitative understanding of their motivations and strategies for coping with the game situations. In debriefing subjects about their game play, many reported spontaneously sending signals of 'safe' or 'danger' even without having received any information. Some said they 'just got nervous and panicked' and thus sent a 'danger' signal without any other reason; and others said that they felt that 'no news is good news', and so they sent a 'safe' signal to their neighbours. Many subjects indicated that they had different thresholds for passing on messages (ranging from just one contact of theirs expressing an opinion about safety or danger, to all four). Many said that they would send a signal (e.g. a 'danger' signal) and then pause for a few seconds to re-send it, so as to be sure that 'others were listening'. Yet, those who were connected to the actual informant (and who, by design, knew this fact) often misinterpreted any such behaviour by the informant as indicating that the informant was unsure of what was happening, and so such individuals sometimes did *not* pass on the (true) informant's message to others. Some subjects reported sending a 'danger' signal and waiting to see if their neighbours copied it before evacuating, and others did not wait for such confirmation, feeling that their job was done when they passed on the message they got. Most subjects indicated that they had higher thresholds for evacuation than for just signalling. In short, players manifested and explained a great variety of behaviours even though they all understood the rules of the game and even though some of these behaviours would not be seen as strictly rational. A wide range of behaviours appeared within our experimental groups, just as in real-world situations [5,34,35].

## (c) Individual behaviours upon exposure to neighbours' signals

How does information signalling from neighbours affect evacuation behaviour in general? To quantify this, we apply the epidemiological concepts of exposure and competing risks [36]. Each subject made decisions regarding sending a signal (safe or danger) and also regarding evacuation (yes or no) while seeing the node colour of four local neighbours indicating safe or

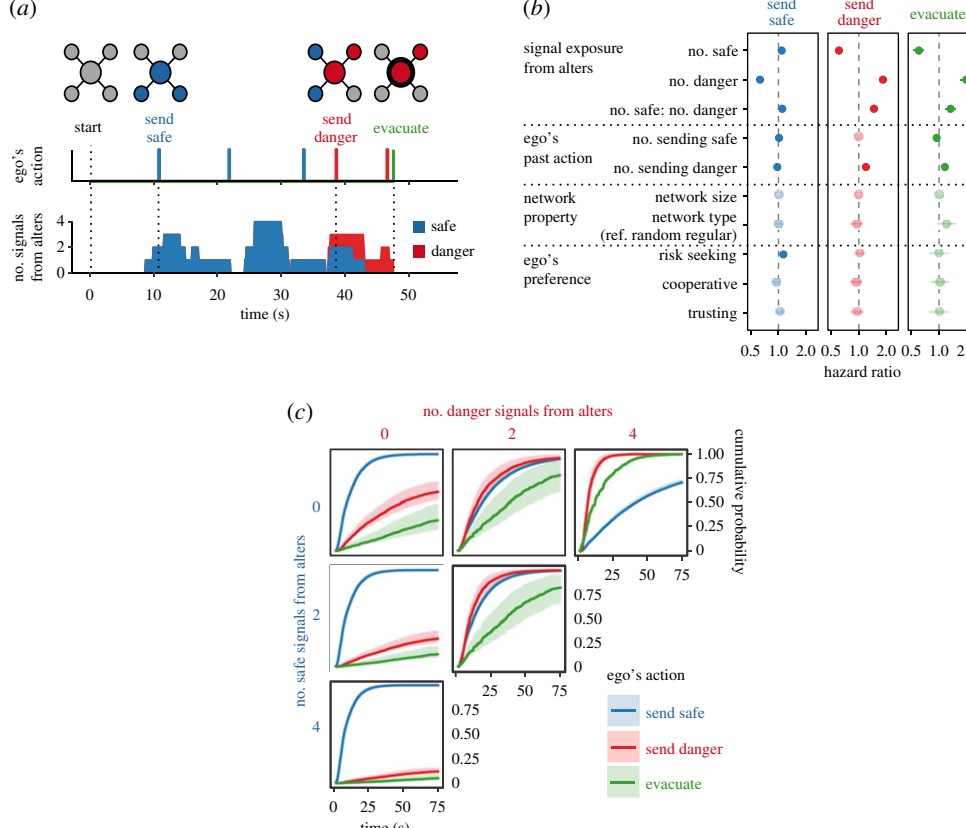

**Figure 4.** Individual behaviours upon exposure to neighbours' signals. (*a*) An example of an individual taking action upon receiving a series of signals from neighbours. The figures are snapshots with the local network of a player when they first send safe signals, send danger signals and eventually evacuate. (*b*) The hazard ratios are estimated by Cox models with time-varying covariates of neighbour's signals, incorporating random effects for individuals. The model includes control variables of subject's past action, network properties and subject's psychological features (see Material and methods for full models and details). Error bars are 95% confidence intervals. Dark-colour plots for significant coefficients with 95% confidential intervals. (*c*) Cumulative probability of sending a safe signal, sending a danger signal and evacuating, over time, while controlling for signal exposure from neighbours, estimated by the model. Error bars are 95% confidence intervals. The curves show the probability of taking the action at a given time if a player receives the indicated signal combination from his or her neighbours. Safe signals in blue, danger signals in red and evacuation in green (see electronic supplementary material, figure S5 for more details). (Online version in colour.)

danger (figure 4*a*). We assume that the signals from neighbours constantly affect the subject's instantaneous risk of various behaviours.

We therefore examined the impact of neighbours' signals on individual behaviours using Cox regression models with time-varying covariates indicating the number of signals in neighbours, incorporating a random effect for individuals (figure 4*b* and electronic supplementary material, table S2). The behavioural model accommodates repeated events of different actions by the individuals by controlling for the number of their past actions (i.e. status quo bias in decision-making [12]; we also confirmed the consistency of results with other model; see electronic supplementary material, table S3). The model also includes global network properties and *ex ante* heterogeneity of individuals as control variables. Figure 4*b* clarifies the impact of signal exposure from neighbours on subjects' decision-making. Every time an individual receives a signal from neighbours, this increases the risk of sending the same signal and reduces the risk of sending the

opposite one. We find that the impact of warnings is larger than that of safe messages. Evacuation behaviour is also affected by signal exposure from neighbours in the same pattern.

Based on the behavioural model, we estimated the cumulative probability of subject's actions over time depending on combinations of signal exposures from neighbours (figure 4c; see electronic supplementary material, figure S5 for the full version). These curves represent the probability of sending a safe signal, sending a danger signal, and evacuating, by the indicated time, depending on whether the subject has been receiving the indicated signal combinations from neighbours since the session began.

In keeping with the in-person debriefings summarized above, figure 4c shows that subjects are more likely to send a safe signal than a danger signal when they did not have *any* signals from neighbours. Once someone starts sending a safe signal, it increases the neighbours' hazard of sending a safe signal and this constrains the impact of warnings and evacuations at both the individual and collective levels. Safe signals discourage the receiver from sending a warning (the hazard decreases 0.61 times for every safe signal) and from evacuating (the hazard also decreases 0.60 times). Thus, these diffusion dynamics favour safe signals more than danger signals (figure 3), and this suppresses even necessary evacuation (figure 2).

Subjects were unaware of the global network structure in which they were embedded because we did not inform them of the network treatment to which they were assigned and the number of neighbours was always the same between the random-regular and the ring-lattice networks and between the small and the large networks. We confirmed with the behavioural model that the effects of network topology and size, which are shown in figure 2, are entirely mediated by individual exposure (figure 4b). Hence, network topology and size alter individual's signal exposure and ultimately evacuation behaviour without subjects realizing it.

We also examined how individual psychology measures are reflected in behaviours in these time-critical situations. Before the game, each subject was asked with a five-level rating system about his or her preferences regarding risk, cooperation and trust, modelled on prior work [31] (see Material and methods). We found that, as shown in the survey of actual disaster victims [5], risk-seeking subjects were less likely to evacuate without network interactions ($p = 0.024$ in subjects of the independent condition; see electronic supplementary material, figure S6), but we also found that individual's risk preference had no effect on evacuation behaviour with network interactions after controlling for their signal exposure ($p = 0.980$ in subjects of the network condition; figure 4b). The results show that subjects rely on the opinions of others instead of following their own inclinations when they are embedded in networks. There were no significant differences with respect to measures of cooperation and trust.

## (d) Further exploration of the effect of post-evacuation information and structural heterogeneity

In a separate, further set of experiments involving an additional 1700 subjects (beyond the 2480 considered above), we explored circumstances that might make networked subjects evacuate more often than isolated ones. We examined two features: post-evacuation information availability and structural heterogeneity (see electronic supplementary material, figure S7). We tested the additional treatments solely with the networks of 20 nodes. These treatments, however, did not make the social response to 'disasters' significantly better than that seen in the independent condition.

First, to further evaluate the impact of information availability, in a variation on our experiments (involving $N = 1280$ subjects in 64 groups), we allowed already-evacuated subjects to send additional signals to their neighbours in two ways (electronic supplementary material, figure S7A). In the 'continuous communication' condition, subjects could use the communication buttons even after they evacuated. As subjects did not need to refrain from evacuation in order to continue to send signals, there was no dilemma between self-preservation and altruism in this condition. In the 'visible evacuation' condition, the nodes of evacuated subjects were removed in

their neighbours' network diagram. Thus, subjects could see what neighbours *did* (i.e. whether they evacuated or not) in addition to what they *signalled* (safe or danger). We found that the additional indication from evacuated subjects increased successful evacuations from 'disaster' in random-regular networks, and it increased false evacuations in cases of no 'disaster' in ring-lattice networks. Even in situations that allow the post-evacuation communication or that make evacuation visible, subjects who exchange information in a network are, overall, still less likely to evacuate, compared to when they make a decision alone (electronic supplementary material, figure S7A).

Second, in another set of experiments, involving 420 subjects in 21 groups, we created small world and random graph networks (i.e. networks with heterogeneous degree); the existence of shortcuts in the networks did not meaningfully increase correct evacuation (see electronic supplementary material, figure S7B).

## (e) Quantifying losses

Finally, we quantified the relative balance of the losses across natural and social causes as a result of the interplay between interpersonal communication and behavioural decisions. In the network of 20 nodes, a subject would earn $3.90 by doing nothing in a situation without 'disaster' (i.e. the maximum possible payoff; table 1). When a 'disaster' struck and all subjects evacuated from a 'disaster', a subject would earn $2.90 by paying the $1.00 evacuation cost. That is, if subjects perfectly communicated with each other, they would lose only $1.00 per person (i.e. the loss due to natural cause). In the actual 'disaster' sessions, the average earning per subject embedded in a network is $0.47. Subjects actually lost $3.43 per person compared to the maximum possible earnings. Thus, the loss due to the miscommunication (the social cause) is $2.43 (=3.43–1.00). Moreover, this 'socially caused loss' increases as network size increases.

## 4. Discussion

In our experiments, interpersonal communication reduced needless evacuations when there was no danger, but, at the same time, it also reduced necessary evacuations when there was a danger— even when a knowledgeable person in the group could correctly announce the existence of an impending 'disaster'. The diffusion of safe messages—often false—frequently overwhelmed the diffusion of warnings—even if true. This self-enforcing norm of a sense of safety spontaneously emerged in almost all 'disaster' sessions, even though subjects understood the rules of the game and even though this behaviour might seem 'irrational'. In the absence of information, people spontaneously generate rumours, and networks can magnify this, especially when the rumours are good news. Larger networks with a smaller proportion of informed subjects suffered more damage due to this human-caused misinformation. People in our experiments often displayed both the procrastination and false reassurance seen in real emergency situations, and normalcy bias magnified the damage from collective dangers [21,22]. In fact, people may be harmed *more* by a collective threat precisely because they can communicate over a network more easily [3], as our calculations also show.

Our work is in keeping with work showing that the wisdom of crowds [37] is optimized when decision-making is independent [15–18]. In this regard, however, there is major difference from prior work [38–40] in that the collective decision-making here decreased both false positives and true positives. The reasonably asymmetric payoffs between the behavioural options here can cause the propagation of false negatives. Given that warnings (bad news) are valuable but undesirable messages, uninformed individuals amplify popular taste and refuse to listen to unpopular opinions from the minority (i.e. the informant) [17]. As a result, social influence favoured only one side—not to evacuate—and thus amplified 'disaster' risk [18].

We find that emergency evacuation (whether warranted or not) in this simplified scenario is actually encouraged by the social reinforcement that is fostered by the network transitivity inherent in lattice networks. The result is consistent with prior experimental and observational

work on behavioural contagion [26,41]. Although theoretical models suggest that low-clustered networks can promote contagion [14,25], such models often consider things that spread via a single contact, such as a germ, in what is known as 'simple contagion'. However, as we show in the individual-level behavioural model (figure 4), people often aggregate information across sources they are in contact with in order to make a decision, especially under uncertainty (a process also seen in other animals [42]). In multiple-contact transmission, redundant ties (connections between neighbours or network transitivity) facilitate the spread of a costly practice (such as evacuation, here) in what is known as 'complex contagion' [26,41,43].

Still, our experiments also show the negative side of the social reinforcement in the welter of opinion, and the limitations of accurate information propagation, as typified by fake-news turmoil [8]. Highly clustered networks, especially when the networks are large, create a sharp conflict of views and make it difficult to form an accurate consensus as a whole group [43]. More generally, social systems where both positive and negative influences exist may exhibit distinctive collective dynamics compared to those in which only positive influences exist, and this may make it more difficult to reach a critical mass for contagion of information or, especially, action. The spread of good and bad news may even be seen as a kind of 'duelling contagion' [2].

Even though the communal dynamics and thresholds for actions that we describe provide mechanistic insights, the specifically psychological mechanisms underlying the observed behaviours were not explored here (beyond our basic assessment of risk-tolerance and trust). Although the expected profit of evacuation was higher than that of staying under an even chance of a 'disaster' (table 1), less than half subjects evacuated in the independent condition (figure 2). This behavioural bias towards staying can perhaps be partly explained as a sort of endowment effect [12]; subjects with an endowment at the outset of a game were unwilling to trade it for their security in conditions of ignorance or uncertainty. Also, subjects might underestimate the 'disaster' risk even though they were informed that 'a disaster may or may not strike' because extraordinary accidents rarely happen in real life. As an alternative explanation, subjects might find it easy to make a risky choice with the such small economic stakes as in our game. Although these explanations are plausible with respect to the behavioural biases in the independent condition, they cannot (given our randomized experimental set-up) fully explain our main finding that the people connected with each other evacuated less than those isolated even when a 'disaster' materialized (figure 2).

Hence, we found a bias towards staying that was compounded in communication networks by the spreading of safe signals more than danger signals (figures 3 and 4c). Why might individuals express safe signals more than danger signals, as a default? Some subjects might start communication with their initial belief of no 'disaster', lacking a long-term horizon (i.e. 'discounting' of future gains [44]; see Supplementary Material). This self-serving explanation is supported by the fact that risk-seeking and selfish subjects were more likely to send safe signals (figure 4b). In addition, some uninformed individuals might use the well-known heuristic of 'no news is good news' [11], as our in-class debriefings also explicitly suggested. They might interpret no signals as a sign of safety and share their view with others. This corresponds to the fact that remarkably low percentages of people evacuated absent any (relevant) signals from network neighbours (figure 4c). In contrast with sending safe signals, some subjects might hesitate to issue warnings because they might be afraid of inducing social anxiety and confusion [45]. Finally, subjects might simply seek an empathetic ear. Quite a few subjects reported having anxiety and stress during gameplay in qualitative comments about their feelings after the experiment was over. They might use safe signals to keep in touch with others. Not only instrumental, but also emotional, tie activation might occur [46].

It is important to note that simple, random play is not compatible with the experimental results. The spreading of safe and danger signals would have occurred at the same frequency if subjects randomly sent signals without understanding the payoff difference. Mere distraction is also not a sufficient reason for staying longer in the network sessions (compared to the individual

ones) because the evacuation rate varies with network topology and size (which subjects could not recognize (figure 2)).

Although, in our supplementary experiments, we explored the effect of additional information and non-uniform local connections (i.e. networks with people having different numbers of connections; see electronic supplementary material, figure S4), there are other features potentially relevant to networked response to collective dangers—for example, one might examine whether multiple informants [17,47], unequal weight or different types of ties [40,48], verbal communication or subjects' previous experiences [5] can affect the outcome. Different payoff structures could also affect our experimental results. However, we confirmed with one supplementary study that an even smaller evacuation cost was sufficient to keep subjects from evacuating in the network condition (electronic supplementary material, figure S8). On the other hand, it seems likely that more serious potential damage from a disaster could promote evacuation in both the independent and the network conditions. Subjects might also change how long a time they spend on information sharing if they were informed about group size or network topology in advance. These are all areas for future work. Another promising topic is the long-term response of social systems to repeated exposures to uncertain dangers from the perspective of an evolving network [49].

Social interactions have complex effects in the uncertain situation created by a possible crisis, promoting the spread of both true and false information regarding both safety and danger, and facilitating both helpful and unhelpful responses. In a sense, interpersonal communications may decrease actual security in return for collective reassurance. Although the results of laboratory experiments do not translate directly into the real world, the evidence presented here suggests that formal details of interpersonal communications might place humans at systematic risk when facing a collective danger [50]. Given the growing dependence of personal communication channels and their widening scale, the negative aspects of network reinforcement may intensify [8]. Humans have an evolved psychology when it comes to responding to collective threats to feel anxiety and fear in isolation, but modern communication technology may provide dangerous and false reassurance [51]. Although social networks excel at providing social support, they may work poorly as information pathways for inconvenient truths, especially when it matters.

# 5. Material and methods

## (a) Experimental design

A total of 4348 subjects ($N = 2648$ for main experiments; $N = 1700$ for supplementary experiments) participated in our incentivized decision-making game experiments. Subjects were recruited using Amazon Mechanical Turk (MTurk) via our breadboard software platform (available at breadboard.yale.edu). MTurk is an online labour market in which employers contract with workers to complete short tasks for relatively small amounts of money. Many studies have demonstrated the validity of behavioural experiment data gathered using MTurk (e.g. [52,53]). Behaviours of MTurk subjects in stylized economic games are correlated with their actual behaviours in a real-world situation [31]. The experiments were conducted from July to September 2016, from January to April 2017 and from February to June 2018. All subjects consented, and this research was approved by the Yale University Committee of the Use of Human Subjects.

To observe the effects of network topology and size while keeping other initial conditions the same, we completed 108 sessions for network conditions with 2480 subjects (48 networks with 10 subjects each, 32 networks with 20 subjects each, 16 networks with 40 subjects each and 12 networks with 60 subjects each; half of the networks were random-regular networks and the other half were ring-lattice networks) and 168 sessions for the independent condition with 168 solitary subjects.

Each session lasted 75 s. In each session, the subjects were paid a $1.00 show-up fee and a bonus depending on whether they took the appropriate decision with respect to an impending disaster. When a disaster struck before they evacuated, the subjects earned no bonus. Otherwise, they earned a bonus of $2.00 without the disaster or $1.00 with the disaster by spending $1.00 to evacuate, plus $0.10 per other player who took the correct action accordingly (table 1). This roughly captures the positive externalities on people who survive a crisis in the real world. For example, when a possible disaster did not materialize, individuals (including those who returned from a false evacuation) reap benefits from economic and social activities, public infrastructure and public safety maintained by people who stayed on-site.

At the start, subjects were required to answer whether they agree or disagree with three sentences to evaluate their personal preference with a five-level rating system (completely agree, agree, neither agree nor disagree, disagree, completely disagree). The sentences were 'I am willing to take risks, in general'; 'People should be willing to help others who are less fortunate; and 'Most people can be trusted'. Prior work shows that responses are correlated with behavioural patterns which appear consistently in the gameplay of several dyadic economic games, such as the dictator game [31].

After answering the survey, each subject was asked to take a tutorial before the actual game would begin. In the tutorial, each subject separately interacted with three dummy players in a 45-second practice game.

After the practice game, subjects were assessed for their comprehension of the game rules and payment structure using three multiple-choice questions, each with three options. If they failed to select the correct answer to any of the questions, they were dropped from the game. At 720 s after the tutorial began, a 'Ready' button became visible simultaneously to all the subjects who completed the tutorial and passed the comprehension tests. The real games started 30 s after the 'Ready' button showed up. If subjects did not click the button before the game started, they were dropped. The games of networked groups required an exact number of subjects. When the subjects who successfully clicked the button were more than the required number, surplus subjects, who were randomly selected, were dropped from the game. When the number of qualified subjects was less than the required number, the game did not start (and subjects got their show-up fee). In total, 41.5% of all participants joined the game ($N = 4348$), 23.7% failed to pass the comprehension tests, 15.5% failed to click the 'Ready' button (including those who stopped in the middle of the tutorial, those who could not complete it before the game started, and those who missed the button within the 30-second time window) and 19.3% were dropped given our specifications regarding group size.

At the start of the game, we selected one subject (the 'informant') at random who was informed in advance whether a disaster would indeed strike or not. The behaviours of the informants did not vary significantly across the network treatments (electronic supplementary material, figure S9). The other subjects were informed that some players had accurate information about the disaster, and they were also told that immediate neighbours of the informant would know the identity of the informant by our marking 'i' on the informant's node in their network diagram in the game screen (see electronic supplementary material). The exact sentence that the informants received in their game screen was 'A disaster is going to strike!' when a disaster would strike or 'There is no disaster'. when a disaster would not strike. Uninformed subjects received notification that 'A disaster may or may not strike'.

Each session ended in 75 s. In half of the sessions, a disaster struck at the end of the game. We did not inform any subjects, including the informants, when their sessions would end, the global network structure in which they were embedded, or how many informants were in the game.

In the supplementary experiments, involving $N = 1280$ subjects in 64 groups, we also manipulated additional information that was apparent from evacuated subjects to their neighbours (see electronic supplementary material, figure S7A). The main experiment setting simulates the situation of people who simply lose communication with each other during an evacuation. That is, once subjects chose to evacuate, they could not use the communication buttons, but their node was left in a network (set to a grey colour). In particular, the neighbours

of evacuated subjects were not informed of their evacuation (they simply observed that the departed node communicated neither safe nor danger signals). In the additional two conditions, in terms of information availability, evacuated subjects could send additional signals to their neighbours in a different way. In the 'continuous communication' condition, subjects could use the communication buttons even *after* they evacuated. As subjects did not need to refrain from evacuation in order to continue to send signals, there was no dilemma between self-preservation and altruism in this condition. In the 'visible evacuation' condition, the nodes of evacuated subjects were removed in their neighbours' network diagram. In contrast with other conditions, subjects could see what neighbours *did* in addition to what they *signalled* (about their sense of whether a disasters was about to strike or not).

We also tested two further types of heterogeneous network structure in additional supplementary experiments involving groups of 20 subjects (see electronic supplementary material, figure S7B), in $N = 420$ subjects in 21 groups. In these further experiments, we created ring-lattice networks with shortcuts; these networks, known as 'small-world' networks [25], were created by rewiring an opposite link pair of a ring-lattice. The others were standard random networks generated by Erdős-Rényi model [54] with the same number of links as the other networks; these random networks had non-uniform local connections (i.e. subjects did not always have four neighbours).

## (b) Modelling of expected profit and whether to evacuate

In the experiment, players receive a bonus based on the combination of their behaviour choice (whether to evacuate) and an environmental risk factor 'disaster' (table 1). When a player elects to evacuate and then a disaster strikes, the player earns \$1.00 (by paying the evacuation cost \$1.00) and \$0.10 for each other player who has successfully evacuated. When a player evacuates but a disaster does not strike, the player earns \$1.00 and \$0.10 for each other player who stays to the end. When a player does not elect to evacuate until the game ends and a disaster does not strike, the player earns \$2.00 and \$0.10 for each other player who also stays to the end. When a player does not evacuate but a disaster strikes, the player earns nothing.

In the independent (solitary) condition where subjects are not connected with each other, they do not know whether a disaster will strike in advance (other than the informants). When such subjects see the chance of a disaster as 50%, their expected profit of each behaviour choice (whether to evacuate) is

$$E_{\text{Evacuate}} = 0.5 \times (1.0 + 0.1 n_{\text{Evacuate}}) + 0.5 \times (1.0 + 0.1 n_{\text{Stay}})$$
$$= 1.0 + 0.05 \times (n_{\text{Evacuate}} + n_{\text{Stay}})$$

and

$$E_{\text{Stay}} = 0.5 \times 0.0 + 0.5 \times (2.0 + 0.1 n_{\text{Stay}})$$
$$= 1.0 + 0.05 \times n_{\text{Stay}},$$

where $n_{\text{Evacuate}}$ is the number of other players who have evacuated before a disaster strikes and $n_{\text{Stay}}$ is the number of other players who stay to the end. Under a purely uncertain condition, the expected profit of evacuation is greater than or equal to that of staying to the end (i.e. $E_{\text{Evacuate}} \geq E_{\text{Stay}}$), regardless of any other players' choice. Only when all other players elect to stay is the expected profit of evacuation equal to that of staying.

## (c) Modelling of evacuation hazard with signal exposure from neighbours

We analysed how individual evacuation behaviour varies with exposure to signals from neighbours [36]. Let

$$a_i^{\text{evacuate}}(t) = \begin{cases} 1 & \text{if subject } i \text{ evacuates at time } t \\ 0 & \text{otherwise} \end{cases},$$

$$a_i^{\text{showsafe}}(t) = \begin{cases} 1 & \text{if subject } i \text{ shows a safe signal at time } t \\ 0 & \text{otherwise} \end{cases},$$

and

$$a_i^{\text{showdanger}}(t) = \begin{cases} 1 & \text{if subject } i \text{ shows a danger signal at time } t \\ 0 & \text{otherwise} \end{cases}$$

The hazard function, or instantaneous rate of occurrence of subject $i$'s evacuation at time $t$, is defined as:

$$\lambda_i(t) = \lim_{dt \to 0} \frac{\Pr(a_i^{\text{evacuate}} = 1; t < T \le t + dt | T > t)}{dt}.$$

To model the time to evacuation, we used a Cox proportional hazards model with time-varying covariates for the number of signals, incorporating an individual actor-specific random effect (figure 4*b*; electronic supplementary material, table S2) [55]

$$\lambda_i\{t | X_i(t), Y_i(t), G_i, P_i\} = \lambda_0(t) \exp\{\beta'_X X_i(t) + \beta'_Y Y_i(t) + \beta'_G G_i + \beta'_P P_i + \gamma_i\},$$

where $\lambda_0(t)$ is a baseline hazard at time $t$.

In the model, the hazard $\lambda_i(t)$ depends on the value of covariates $X_i(t)$, $Y_i(t)$, $G_i$, and $P_i$. The covariate $X_i(t)$ is the vector of the number of safe signals $x_i^{\text{safe}}(t)$, the number of danger signals $x_i^{\text{danger}}(t)$ and their interaction in the neighbours of subject $i$ at time $t$. When subject $j$ is a neighbour of subject $i$ (i.e. $j \in N_i$), subject $i$ is exposed to the signal of subject $j$, so that

$$x_i^{\text{safe}}(t) = \sum_{j \in N_i} a_j^{\text{show safe}}(t)$$

and

$$x_i^{\text{danger}}(t) = \sum_{j \in N_i} a_j^{\text{show danger}}(t).$$

This modelling shows how the hazard of individual's evacuation depends on the signalling actions of others through the network.

The covariate $Y_i(t)$ is the vector of number of the subject $i$'s actions of sending safe and danger signals before time $t$. The covariate $G_i$ is the vector of the global properties of the network in which subject $i$ is embedded, a topology indicator and a network size indicator. The covariate $P_i$ is the vector of subject $i$'s survey scores regarding personal preferences about risk, cooperation and trust. The coefficients $\beta$ are the fixed effects and $\gamma_i$ is the random effect for individual $i$. We assumed that waiting times to evacuation in different actors are conditionally independent given the sequence of signals they receive from network neighbours.

We included global network properties $G_i$ and subject's personal preference $P_i$ as control variables. Under the hypothesis that individuals only act in response to their own environment (neighbourhood) signals, it is unlikely that the network topology and size directly affects individual's decision-making because subjects had no means of knowing global network properties. Subjects were not informed of them in advance and they always had four neighbours in all the sessions of the key social network conditions. If the effects of network type and size are entirely mediated by individual exposures, the coefficients corresponding to $G_i$ will be estimated to be zero.

The measures of personal preference $P_i$ are the result of prior questionnaires in which subjects reported their level of agreement or disagreement for a series of statement with a five-level rating

system (Do you agree or disagree with following statement?': 2 for 'Completely agree', 1 for 'Agree', 0 for 'Neither to agree nor disagree', −1 for 'Disagree', and −2 for 'Completely disagree') [31]. There was no signal exposure without network structure in the independent condition. Thus, the behavioural model of the independent condition has only personal preference as covariate (see electronic supplementary material, figure S6), so that

$$\lambda_i\{t|P_i\} = \lambda_0(t)\exp\{\beta'_P P_i\}.$$

We also applied the same model to the signalling behaviour. In contrast with evacuation, which can only happen once, subjects can send a danger or safe signal several times. An assumption of independence between repeated actions of the same subject may be suspect, so we included a count of the number of previous signals (safe/danger) $Y_i(t)$ in the behavioural model. To test the robustness of the conditional model, we also applied a counting process model with the assumption of independence between repeated actions. We found that the conditional model fits the data significantly better than the independent model ($p < 0.01$ for all the models of evacuation, sending a safe signal and sending a danger signal; likelihood ratio test). Also, the estimated coefficients of the independent model are almost the same as those of the conditional model (electronic supplementary material, table S3).

## (d) Analysis of diffusion speed of signals

We identified the subjects who sent a signal when their neighbours had never sent one as (spontaneous) 'diffusion sources'. When a subject sent a signal after at least one neighbour had sent the same type of signal, we regarded the subject's signalling as occurring in a chain of signal diffusion and calculated the time elapsed after the chain's source sent the initial signal.

Using this procedure, we found that, while safe signals spread significantly faster than danger signals ($p < 0.01$; Kolmogorov–Smirnov test), the diffusion speed has no meaningful difference with respect to the signal's veracity ($p = 0.26$; Kolmogorov–Smirnov test). As a result, false reassurances spread faster than true warnings (figure 3c). Subjects are more likely to send a safe signal than to send a danger signal throughout the game, even when a 'disaster' is to strike and an informant can circulate an accurate alert of this fact.

Data accessibility. The data in this manuscript are available at the Human Nature Laboratory Data Archive (http://humannaturelab.net/publications/collective-communication-and-behaviour-in-response-to-uncertain-danger-in-network-experiments).
Authors' contributions. H.S. and N.A.C. designed the project. H.S. collected the data and performed the statistical calculations. H.S., F.W.C. and N.A.C. analysed the results. H.S., F.W.C. and N.A.C. wrote the manuscript.
Competing interests. The authors declare no competing interests.
Funding. Support was provided by grants from the Robert Wood Johnson Foundation, Tata Sons Limited, Tata Consultancy Services Limited, Tata Chemicals Limited, the Nomis foundation, and the National Institute of Social Sciences. F.W.C. was funded by NIH grant no. NICHD 1DP2HD091799-01.
Acknowledgements. We thank M. Salganik, E. Erikson, J. Farrell, A. Almaatouq and J. Becker for comments. M. McKnight provided programming needed for the online experiments, and M. Kawakatsu, J. Cutler-Tietjen and K. Garcia provided assistance running the experiments.

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
