## [Reviewer comments · Proceedings. Mathematical, Physical, and Engineering Sciences]

Review History

RSPA-2019-0685.R0 (Original submission)

Review form: Referee 1

Is the manuscript an original and important contribution to its field?

Acceptable

Is the paper of sufficient general interest?

Acceptable

Is the overall quality of the paper suitable?

Good

Can the paper be shortened without overall detriment to the main message?

Yes

Do you think some of the material would be more appropriate as an electronic appendix?

No

Do you have any ethical concerns with this paper?

No

Recommendation?

Major revision is needed (please make suggestions in comments)

Comments to the Author(s)

Dear authors and editor,

The authors present experimental results on how information about a fictitious disaster (or its absence) spreads on a variety of communication networks.

The paper presents a thorough analysis of an abstract experimental system and is, as far as I can tell, methodologically sound.

As the authors acknowledge (although perhaps not sufficiently explicitly), the experimental setup has some limitations in that it only considers a highly constrained aspect of how humans communicate. Specifically, the following aspects are likely to be relevant for the observed behaviours and I would recommend the authors highlight these points in more detail at earlier in their manuscript:

- the role of additional, independently verifiable information. At present the only information available to most individuals is what others communicate.
- the role of extending the content of messages. For example, including a "I don't know" message may help to satisfy individuals' need to communicate for re-assurance that the authors mention in the discussion.
- type of social contacts. The author mention link weightings, but relationship status could be relevant (e.g. would you trust you friends' messages more?)
- number of informed individuals. While the authors vary the percentage of informed individuals in their networks, they only inform one individual. Thus, much rests on the behaviour of one individual. Some data on what the informed individual did across experiments would also be useful (how many messages did they send? How frequently did they evacuate successfully and when?).

In addition to these points, I have several more specific comments that I list below. To summarise, I think this is a sound study that investigates one specific niche case of how humans communicate.

I hope you find my comments useful and constructive.

****Specific comments****

In general, I'm not convinced having a separate methods section at the end of the manuscript is useful. I found myself having to jump between sections to follow what exactly you were doing in experiments and additionally, quite a bit of information is duplicated across the main text and the methods section. I'd recommend you have a dedicated methods section earlier in the main text.

Keywords and manuscript: this is probably a question of definitions, but I think it is a bit of a stretch to refer to your networks as social networks. To me, they seem to be communication networks, considering the lack of meaning and weight attached to links.

Will the data be made publicly available? You only state it will be stored at a permanent URL. An official data repository would be preferable.

Page 3 (p. number submission system), line 10-11: while popular, the words panic and stampede are somewhat imprecise and I would encourage the authors to think about alternatives (see e.g. <https://www.hindawi.com/journals/jat/2019/9267643/abs/>)

Some additional literature that may be of interest (role of network structure on group decision-making):

<https://royalsocietypublishing.org/doi/full/10.1098/rsif.2015.0213>

<https://link.springer.com/article/10.1007/s00265-012-1331-6>

<https://www.sciencedirect.com/science/article/pii/S0968090X18307009>

Payoff: with your payoff structure, would the expected payoff be higher for larger groups under the assumption of random behaviour? Could that affect your results?

What would you expect to happen if the payoff matrix was altered (e.g. higher penalties for not evacuating, higher social incentive...?).

Page 11, lines 2-4: please report the value of the test statistic and the precise p-value for all statistical tests.

Page 14, lines 14-56: while interesting, it is not clear how this information was elicited from people. Just by asking them individually or in front of the others? How many people said what? Social influence you affect what people told you.

Page 15, lines 5-10: you claim that you observed a 'very realistic range of behaviours'. Is there any evidence for supporting this statement? I.e. have similar behaviours been previously observed in real emergencies...?

Review form: Referee 2

Is the manuscript an original and important contribution to its field?

Good

Is the paper of sufficient general interest?

Good

Is the overall quality of the paper suitable?

Excellent

Can the paper be shortened without overall detriment to the main message?

Yes

Do you think some of the material would be more appropriate as an electronic appendix?

No

Do you have any ethical concerns with this paper?

No

Recommendation?

Accept with minor revision (please list in comments)

Comments to the Author(s)

The manuscript 'Collective Communication and Behaviour in Response to Uncertain "Danger" in Network Experiments' is a very interesting interdisciplinary contribution sitting at the interface of several fields, including game theory, network science and human behavior. In this work, through a series of experiments the authors simulate situations where an imminent disaster might not or might be imminent, with this information only given to a single initial individual, who is the seed of the subsequent process of information spreading. The authors test different network structures, and compare them to a controlled scenario with absence of social network. In addition to what is presented in the main manuscript, the effect of several interesting network and game features is analysed in the Supplementary material. Results highlight the frequent poor ability of social networks to act as efficient pathways for the spreading of information in the case of disasters, where limited time is allowed for both individual and collective decision making.

The experimental evidence is intriguing, the analyses are rigorous, and the paper is definitely well written. Please find below a list of minor comments before I can recommend publication in RSPA.

1) Payoffs are designed to incentive individuals to stay due to the evacuation cost, but also to be correct. How was the relative contribution of the two terms established? A remark I have is that the payoff do not seem to be normalised as a function of the number of players, making the relative contributions of the two terms have a different weight as a function of the system size. By no means I am asking to add a further experiment, I think the authors have already done a great job at investigating multiple different scenarios and cases. I am just asking if they could comment on this, and possibly address this issue in the manuscript, in the case they believe this might have an impact on their findings (in particular those as a function of the number of players in the experiment).

2) It is explained from the beginning in much detail what the spread of true warnings means. However, in my opinion, highlighting earlier in the manuscript what the spread of misinformation exactly means in the considered experimental set-up would also be beneficial to the reader.

3) Page 17 (15 according to the article numbering, at the bottom). 'Local topology is identical'. More often than not local topology refers also to the information about links between neighbours of an individual, making different the case where there is clustering from the one where it lacks. I am aware the term carries some ambiguity, but for the sake of clarity I think it would be preferable to just say that the number of neighbours is the same.

Decision letter (RSPA-2019-0685.R0)

04-Mar-2020

Dear Professor Shirado

The Editor of Proceedings A has now received comments from referees on the above paper and would like you to revise it in accordance with their suggestions which can be found below (not including confidential reports to the Editor).

Please submit a copy of your revised paper within four weeks - if we do not hear from you within this time then it will be assumed that the paper has been withdrawn. In exceptional circumstances, extensions may be possible if agreed with the Editorial Office in advance.

Please note that it is the editorial policy of Proceedings A to offer authors one round of revision in which to address changes requested by referees. If the revisions are not considered satisfactory by the Editor, then the paper will be rejected, and not considered further for publication by the journal. In the event that the author chooses not to address a referee's comments, and no scientific justification is included in their cover letter for this omission, it is at the discretion of the Editor whether to continue considering the manuscript.

- Acknowledgements
- Funding statement

To revise your manuscript, log into <http://mc.manuscriptcentral.com/prsa> and enter your Author Centre, where you will find your manuscript title listed under "Manuscripts with Decisions." Under "Actions," click on "Create a Revision." Your manuscript number has been appended to denote a revision.

You will be unable to make your revisions on the originally submitted version of the manuscript. Instead, revise your manuscript and upload a new version through your Author Centre.

When submitting your revised manuscript, you will be able to respond to the comments made by the referee(s) and upload a file "Response to Referees" in "Section 6 - File Upload". Please use this to document how you have responded to the comments, and the adjustments you have made. In order to expedite the processing of the revised manuscript, please be as specific as possible in your response to the referee(s).

IMPORTANT: Your original files are available to you when you upload your revised manuscript. Please delete any unnecessary previous files before uploading your revised version.

When revising your paper please ensure that it remains under 28 pages long. In addition, any pages over 20 will be subject to a charge (£150 + VAT (where applicable) per page). Your paper has been ESTIMATED to be 20 pages.

Once again, thank you for submitting your manuscript to Proc. R. Soc. A and I look forward to receiving your revision. If you have any questions at all, please do not hesitate to get in touch.

Yours sincerely
Raminder Shergill
proceedingsa@royalsociety.org

Reviewer(s)' Comments to Author:

Referee: 1

Comments to the Author(s)
Dear authors and editor,

The authors present experimental results on how information about a fictitious disaster (or its absence) spreads on a variety of communication networks.

The paper presents a thorough analysis of an abstract experimental system and is, as far as I can tell, methodologically sound.

As the authors acknowledge (although perhaps not sufficiently explicitly), the experimental setup has some limitations in that it only considers a highly constrained aspect of how humans communicate. Specifically, the following aspects are likely to be relevant for the observed behaviours and I would recommend the authors highlight these points in more detail at earlier in their manuscript:

- the role of additional, independently verifiable information. At present the only information available to most individuals is what others communicate.

- the role of extending the content of messages. For example, including a "I don't know" message may help to satisfy individuals' need to communicate for re-assurance that the authors mention in the discussion.

- type of social contacts. The author mention link weightings, but relationship status could be relevant (e.g. would you trust you friends' messages more?)

- number of informed individuals. While the authors vary the percentage of informed individuals in their networks, they only inform one individual. Thus, much rests on the behaviour of one individual. Some data on what the informed individual did across experiments would also be

useful (how many messages did they send? How frequently did they evacuate successfully and when?).

In addition to these points, I have several more specific comments that I list below. To summarise, I think this is a sound study that investigates one specific niche case of how humans communicate.

I hope you find my comments useful and constructive.

****Specific comments****

In general, I'm not convinced having a separate methods section at the end of the manuscript is useful. I found myself having to jump between sections to follow what exactly you were doing in experiments and additionally, quite a bit of information is duplicated across the main text and the methods section. I'd recommend you have a dedicated methods section earlier in the main text.

Keywords and manuscript: this is probably a question of definitions, but I think it is a bit of a stretch to refer to your networks as social networks. To me, they seem to be communication networks, considering the lack of meaning and weight attached to links.

Will the data be made publicly available? You only state it will be stored at a permanent URL. An official data repository would be preferable.

Page 3 (p. number submission system), line 10-11: while popular, the words panic and stampede are somewhat imprecise and I would encourage the authors to think about alternatives (see e.g. <https://www.hindawi.com/journals/jat/2019/9267643/abs/>)

Some additional literature that may be of interest (role of network structure on group decision-making):

<https://royalsocietypublishing.org/doi/full/10.1098/rsif.2015.0213>

<https://link.springer.com/article/10.1007/s00265-012-1331-6>

<https://www.sciencedirect.com/science/article/pii/S0968090X18307009>

Payoff: with your payoff structure, would the expected payoff be higher for larger groups under the assumption of random behaviour? Could that affect your results?

What would you expect to happen if the payoff matrix was altered (e.g. higher penalties for not evacuating, higher social incentive...?).

Page 11, lines 2-4: please report the value of the test statistic and the precise p-value for all statistical tests.

Page 14, lines 14-56: while interesting, it is not clear how this information was elicited from people. Just by asking them individually or in front of the others? How many people said what? Social influence you affect what people told you.

Page 15, lines 5-10: you claim that you observed a 'very realistic range of behaviours'. Is there any evidence for supporting this statement? I.e. have similar behaviours been previously observed in real emergencies...?

Referee: 2

Comments to the Author(s)

The manuscript 'Collective Communication and Behaviour in Response to Uncertain "Danger" in Network Experiments' is a very interesting interdisciplinary contribution sitting at the interface of several fields, including game theory, network science and human behavior. In this work, through a series of experiments the authors simulate situations where an imminent disaster

might not or might be imminent, with this information only given to a single initial individual, who is the seed of the subsequent process of information spreading. The authors test different network structures, and compare them to a controlled scenario with absence of social network. In addition to what is presented in the main manuscript, the effect of several interesting network and game features is analysed in the Supplementary material. Results highlight the frequent poor ability of social networks to act as efficient pathways for the spreading of information in the case of disasters, where limited time is allowed for both individual and collective decision making. The experimental evidence is intriguing, the analyses are rigorous, and the paper is definitely well written. Please find below a list of minor comments before I can recommend publication in RSPA.

1) Payoffs are designed to incentive individuals to stay due to the evacuation cost, but also to be correct. How was the relative contribution of the two terms established? A remark I have is that the payoff do not seem to be normalised as a function of the number of players, making the relative contributions of the two terms have a different weight as a function of the system size. By no means I am asking to add a further experiment, I think the authors have already done a great job at investigating multiple different scenarios and cases. I am just asking if they could comment on this, and possibly address this issue in the manuscript, in the case they believe this might have an impact on their findings (in particular those as a function of the number of players in the experiment).

2) It is explained from the beginning in much detail what the spread of true warnings means. However, in my opinion, highlighting earlier in the manuscript what the spread of misinformation exactly means in the considered experimental set-up would also be beneficial to the reader.

3) Page 17 (15 according to the article numbering, at the bottom). 'Local topology is identical'. More often than not local topology refers also to the information about links between neighbours of an individual, making different the case where there is clustering from the one where it lacks. I am aware the term carries some ambiguity, but for the sake of clarity I think it would be preferable to just say that the number of neighbours is the same.

Board Member:

Comments to Author(s):

Valid points are raised by both reviewers, especially about payoff, where it would be beneficial to have discussion about how the current structure of payoff influences the results (what would happen if normalised or changed)

Author's Response to Decision Letter for (RSPA-2019-0685.R0)

See Appendix A.

RSPA-2019-0685.R1 (Revision)

Review form: Referee 1

Is the manuscript an original and important contribution to its field?

Good

Is the paper of sufficient general interest?

Good

Is the overall quality of the paper suitable?

Good

Can the paper be shortened without overall detriment to the main message?

Yes

Do you think some of the material would be more appropriate as an electronic appendix?

No

Do you have any ethical concerns with this paper?

Yes

Recommendation?

Accept with minor revision (please list in comments)

Comments to the Author(s)

The authors address most of my comments.

Minor points remaining:

I previously suggested that the term social network is not appropriate for describing the communication networks the authors implement. The authors have changed the keyword, but continue to refer to 'social networks' instead of 'communication networks' throughout the manuscript.

page 3, line 26 and page 22, line 40: still contain the word panic, even though it is imprecise.

Review form: Referee 2

Is the manuscript an original and important contribution to its field?

Good

Is the paper of sufficient general interest?

Good

Is the overall quality of the paper suitable?

Good

Can the paper be shortened without overall detriment to the main message?

Yes

Do you think some of the material would be more appropriate as an electronic appendix?

No

Do you have any ethical concerns with this paper?

No

Recommendation?

Accept as is

Comments to the Author(s)

I enjoyed reading this revised version of the manuscript. I am broadly satisfied with how the authors addressed my previous comments, as well as those of the other referee. I can now recommend acceptance in Proceedings A.

Decision letter (RSPA-2019-0685.R1)

Dear Professor Shirado

On behalf of the Editor, I am pleased to inform you that your manuscript entitled "Collective Communication and Behaviour in Response to Uncertain "Danger" in Network Experiments" has been accepted in its final form for publication in Proceedings A.

Our Production Office will be in contact with you in due course. You can expect to receive a proof of your article soon. Please contact the office to let us know if you are likely to be away from e-mail in the near future. If you do not notify us and comments are not received within 5 days of sending the proof, we may publish the paper as it stands.

Open access

You are invited to opt for open access, our author pays publishing model. Payment of open access fees will enable your article to be made freely available via the Royal Society website as soon as it is ready for publication. For more information about open access please visit http://royalsocietypublishing.org/site/authors/open_access.xhtml. The open access fee for this journal is £1700/\$2380/€2040 per article. VAT will be charged where applicable.

Note that if you have opted for open access then payment will be required before the article is published – payment instructions will follow shortly. If you wish to opt for open access then please inform the editorial office (proceedingsa@royalsociety.org) as soon as possible.

Your article has been estimated as being 20 pages long. Our Production Office will inform you of the exact length at the proof stage.

Proceedings A levies charges for articles which exceed 20 printed pages. (based upon approximately 540 words or 2 figures per page). Articles exceeding this limit will incur page charges of £150 per page or part page, plus VAT (where applicable).

Under the terms of our licence to publish you may post the author generated postprint (ie. your accepted version not the final typeset version) of your manuscript at any time and this can be made freely available. Postprints can be deposited on a personal or institutional website, or a recognised server/repository. Please note however, that the reporting of postprints is subject to a media embargo, and that the status the manuscript should be made clear. Upon publication of the definitive version on the publisher's site, full details and a link should be added.

You can cite the article in advance of publication using its DOI. The DOI will take the form: 10.1098/rspa.XXXX.YYYY, where XXXX and YYYY are the last 8 digits of your manuscript number (eg. if your manuscript number is RSPA-2017-1234 the DOI would be 10.1098/rspa.2017.1234).

For tips on promoting your accepted paper see our blog post:

<https://blogs.royalsociety.org/publishing/promoting-your-latest-paper-and-tracking-your-results/>

Thank you for your submission. On behalf of the Editors of the journal, we look forward to your continued contributions to the Journal.

Best wishes

Raminder Shergill,
Proceedings A Editorial Office
proceedingsa@royalsociety.org

Reviewer(s)' Comments to Author:

Referee: 1

Comments to the Author(s)
The authors address most of my comments.

Minor points remaining:

I previously suggested that the term social network is not appropriate for describing the communication networks the authors implement. The authors have changed the keyword, but continue to refer to 'social networks' instead of 'communication networks' throughout the manuscript.

page 3, line 26 and page 22, line 40: still contain the word panic, even though it is imprecise.

Referee: 2

Comments to the Author(s)
I enjoyed reading this revised version of the manuscript. I am broadly satisfied with how the authors addressed my previous comments, as well as those of the other referee. I can now recommend acceptance in Proceedings A.

Board member comments to Author(s)

The reviewers are satisfied with the revisions authors made. Still, authors should ensure that R1's comment about "communication network" and imprecise term "panic", are addressed with the proof corrections.

Appendix A

Reviewer(s)' Comments to Author:

Referee: 1

Comments to the Author(s)

Dear authors and editor,

The authors present experimental results on how information about a fictitious disaster (or its absence) spreads on a variety of communication networks.

The paper presents a thorough analysis of an abstract experimental system and is, as far as I can tell, methodologically sound.

As the authors acknowledge (although perhaps not sufficiently explicitly), the experimental setup has some limitations in that it only considers a highly constrained aspect of how humans communicate. Specifically, the following aspects are likely to be relevant for the observed behaviours and I would recommend the authors highlight these points in more detail at earlier in their manuscript:

- the role of additional, independently verifiable information. At present the only information available to most individuals is what others communicate.
- the role of extending the content of messages. For example, including a “I don’t know” message may help to satisfy individuals’ need to communicate for re-assurance that the authors mention in the discussion.
- type of social contacts. The author mention link weightings, but relationship status could be relevant (e.g. would you trust you friends’ messages more?)
- number of informed individuals. While the authors vary the percentage of informed individuals in their networks, they only inform one individual. Thus, much rests on the behaviour of one individual. Some data on what the informed individual did across experiments would also be useful (how many messages did they send? How frequently did they evacuate successfully and when?).

--- We thank R1 for the detailed and helpful suggestions, all of which we have tried to address.

We agree with R1 that our experiments have limitations. We understand that other factors can affect the outcome, but it was not feasible to explore the entire parameter space. We agree that information verification, more complex communication, and connection type changes could modify the dynamics in unpredictable ways. We now mention this in the Discussion.

Regarding the concern related to the informed individuals, we have confirmed with further analyses that the nature and quantity of signals sent by informants did not vary significantly across the network treatments. We have added a description of this in the Methods section, and the results are in Figure S9.

In addition to these points, I have several more specific comments that I list below. To summarise, I think this is a sound study that investigates one specific niche case of how humans

communicate.

I hope you find my comments useful and constructive.

****Specific comments****

In general, I'm not convinced having a separate methods section at the end of the manuscript is useful. I found myself having to jump between sections to follow what exactly you were doing in experiments and additionally, quite a bit of information is duplicated across the main text and the methods section. I'd recommend you have a dedicated methods section earlier in the main text.

--- We understand R1's preference, but we followed the author guide of Royal Society and put the Method section after the main text. We tried to make the main text have concise but sufficient information about our experimental setup for readers to understand the main results. We then elaborated the details for those who want to review our study techniques in the Method section. If we should employ a different official format, we would gladly do so.

Keywords and manuscript: this is probably a question of definitions, but I think it is a bit of a stretch to refer to your networks as social networks. To me, they seem to be communication networks, considering the lack of meaning and weight attached to links.

--- We changed the keyword accordingly. Thank you.

Will the data be made publicly available? You only state it will be stored at a permanent URL. An official data repository would be preferable.

--- Yes, the data will be made publicly available at <http://humannaturelab.net/publications/collective-communication-and-behaviour-in-response-to-uncertain-danger-in-network-experiments>. We now describe the URL in the Data availability section.

Page 3 (p. number submission system), line 10-11: while popular, the words panic and stampede are somewhat imprecise and I would encourage the authors to think about alternatives (see e.g. <https://www.hindawi.com/journals/jat/2019/9267643/abs/>)

--- Thank you for the helpful suggestion. We changed the introduction so as to avoid using the words "panic" and "stampede."

Some additional literature that may be of interest (role of network structure on group decision-making):

<https://royalsocietypublishing.org/doi/full/10.1098/rsif.2015.0213>

<https://link.springer.com/article/10.1007/s00265-012-1331-6>

<https://www.sciencedirect.com/science/article/pii/S0968090X18307009>

--- These are useful papers. We now cite them.

Payoff: with your payoff structure, would the expected payoff be higher for larger groups under the assumption of random behaviour? Could that affect your results?

What would you expect to happen if the payoff matrix was altered (e.g. higher penalties for not evacuating, higher social incentive...?).

--- We did not inform subjects of the group size and network topology before the game started. Thus, from the subject's point of view, the expected payoff should not vary with the group size (and network topology). We confirmed with individual-level analyses that subjects did not change their evacuation and signalling behavior based on the group size itself (see Fig. 4B). We have further highlighted this point in the Experiment setup and Discussion sections.

We quite agree that different pay-off structures can also affect our experimental results. We confirmed with a supplementary test that even a smaller evacuation cost was sufficient to keep subjects from evacuating in the social network condition. And we have now added Figure S8 to report these results regarding a different payoff matrix. Of course, more serious potential damage from a disaster can promote evacuation in both the independent and the social network conditions. We have stressed this point in the Discussion section.

Page 11, lines 2-4: please report the value of the test statistic and the precise p-value for all statistical tests.

--- We have reported the statistical test name and precise p-values for all the comparisons in the next paragraphs.

Page 14, lines 14-56: while interesting, it is not clear how this information was elicited from people. Just by asking them individually or in front of the others? How many people said what? Social influence you affect what people told you.

--- We used focus group methods for this. The 160 people participated in groups of 20 to 40, playing the game together and then being collectively debriefed (i.e., in front of one another, as is standard with this approach). We assessed qualitative responses to the game, and did not precisely quantify remarks made by the subjects. We have added more information about our procedure to the manuscript.

Page 15, lines 5-10: you claim that you observed a 'very realistic range of behaviours'. Is there

any evidence for supporting this statement? I.e. have similar behaviours been previously observed in real emergencies...?

--- We have changed the wording of this a bit to better reflect what we meant, and we have also added three citations regarding prior responses to online and offline disasters.

Referee: 2

Comments to the Author(s)

The manuscript 'Collective Communication and Behaviour in Response to Uncertain "Danger" in Network Experiments' is a very interesting interdisciplinary contribution sitting at the interface of several fields, including game theory, network science and human behavior. In this work, through a series of experiments the authors simulate situations where an imminent disaster might not or might be imminent, with this information only given to a single initial individual, who is the seed of the subsequent process of information spreading. The authors test different network structures, and compare them to a controlled scenario with absence of social network. In addition to what is presented in the main manuscript, the effect of several interesting network and game features is analysed in the Supplementary material. Results highlight the frequent poor ability of social networks to act as efficient pathways for the spreading of information in the case of disasters, where limited time is allowed for both individual and collective decision making. The experimental evidence is intriguing, the analyses are rigorous, and the paper is definitely well written. Please find below a list of minor comments before I can recommend publication in RSPA.

--- We are grateful for these positive comments, and for the detailed and helpful suggestions below. We have tried to address all these suggestions.

1) Payoffs are designed to incentive individuals to stay due to the evacuation cost, but also to be correct. How was the relative contribution of the two terms established? A remark I have is that the payoff do not seem to be normalised as a function of the number of players, making the relative contributions of the two terms have a different weight as a function of the system size. By no means I am asking to add a further experiment, I think the authors have already done a great job at investigating multiple different scenarios and cases. I am just asking if they could comment on this, and possibly address this issue in the manuscript, in the case they believe this might have an impact on their findings (in particular those as a function of the number of players in the experiment).

--- We did not inform subjects of the group size and network topology before the game started. Thus, from the subject's point of view, the expected payoff should not vary with the group size. We confirmed with the individual-level analysis that subjects did not change their evacuation and signaling behavior based on the group size itself (see Fig. 4B). We have highlighted this point in the Experiment setup and Discussion sections.

We agree that different pay-off structures can also affect our experimental results. Although we confirmed with a supplementary test that an even smaller evacuation cost was sufficient to keep subjects from evacuating in the social network condition (we have now added Figure S8 regarding this), more serious potential damage from a disaster can promote evacuation in both the independent and the social network conditions. We have stressed this point more in the Discussion.

2) It is explained from the beginning in much detail what the spread of true warnings means. However, in my opinion, highlighting earlier in the manuscript what the spread of misinformation exactly means in the considered experimental set-up would also be beneficial to the reader.

--- Thank you for the helpful suggestion. We have updated the introduction to mention the potential harm of misinformation.

3) Page 17 (15 according to the article numbering, at the bottom). ‘Local topology is identical’. More often than not local topology refers also to the information about links between neighbours of an individual, making different the case where there is clustering from the one where it lacks. I am aware the term carries some ambiguity, but for the sake of clarity I think it would be preferable to just say that the number of neighbours is the same.

--- We agree with R2 and have changed the word to “the number of neighbours.”

Board Member:

Comments to Author(s):

Valid points are raised by both reviewers, especially about payoff, where it would be beneficial to have discussion about how the current structure of payoff influences the results (what would happen if normalised or changed)

--- As we wrote in the response to R1 and R2, we have clarified our experimental setup regarding the information of group size. We also have elaborated the potential impact of different payoff structures in the Discussion and added new figures to the online supplement.

We thank R1 and R2 for the generally favorable comments. We have tried to respond comprehensively to all the comments and to improve the paper accordingly.